# A New Strain of *Metarhizium robertsii* Isolated from Loess Plateau and Its Virulence and Pathological Characteristics against *Monochamus alternatus*

**DOI:** 10.3390/microorganisms12030514

**Published:** 2024-03-03

**Authors:** Ji-Yang Zheng, He-Liang Shi, Dun Wang

**Affiliations:** 1Key Laboratory of Plant Protection Resources and Pest Management of Ministry of Education, College of Plant Protection, Northwest A&F University, Yangling 712100, China; beaubois@foxmail.com; 2Key Laboratory of Integrated Pest Management on the Loess Plateau of Ministry of Agriculture and Rural Affairs, Northwest A&F University, Yangling 712100, China; shl514539978@foxmail.com

**Keywords:** *Metarhizium robertsii*, *Monochamus alternatus*, virulence, symptom observation, histopathological observation

## Abstract

*Monochamus alternatus* is a serious trunk-boring pest. The isolation and utilization of entomopathogenic fungi to manage *M. alternatus* is important. Here, a new strain GQH6 of *Metarhizium robertsii,* isolated from the Loess Plateau, was identified morphologically and molecularly. The virulence of the strain GQH6 against the third-instar larvae of *M. alternatus* was studied. Then, the pathological process, including symptom observation and histopathological observation, was also researched. The corrected mortality was 100% at 10^9^ and 10^8^ conidia/mL, and 88.89 ± 5.88% at 10^7^ conidia/mL. The LC50 was 1.93 × 10^6^ conidia/mL and the LC90 was 1.35 × 10^7^ conidia/mL. And the LT50 of the strain GQH6 was 3.96 days at 10^9^ conidia/mL, and 4.99 days at 10^8^ conidia/mL. These virulence indices showed high virulence against *M. alternatus* larvae. In addition, the symptoms of the infected *M. alternatus* larvae were obvious. After one day, dark spots appeared and increased in number. By four days, white mycelia appeared. Finally, the larvae body became green. Similarly, the histopathological changes after infection were obvious, mainly manifested in muscle tissue rupture, adipose tissue fracture and midgut disintegration. These results demonstrated that the *M. robertsii* strain GQH6 isolated from the Loess Plateau was highly virulent against *M. alternatus* larvae of the third instar.

## 1. Introduction

Entomopathogenic fungi are a class of microorganisms that specifically infect insects, and they regulate many pests in natural ecosystems effectively in an ecology-friendly manner [1]. The utilization of entomopathogenic fungi to manage agricultural and forestry pests has become a new trend in biological pest control [2]. And the important thing is that entomopathogenic fungi can be isolated from nature [3,4]. So entomopathogenic fungi are easier to obtain. Entomopathogenic fungi have the advantages of a broad spectrum, species diversity, and relative safety to humans and other non-target organisms [5]. So fungal pesticides have the ability to become biological alternatives to chemical pesticides [6]. At present, *Beauveria*, *Metarhizium*, *Cordyceps* have been researched deeply [7]. Among them, *Metarhizium*, as one of extensively studied and applied entomopathogenic fungi, has been used to control multiple pests in multifarious environments [8,9]. *Metarhizium robertsii* is a typical entomopathogenic fungus. And some strains of *M. robertsii* have been developed as environmentally friendly fungal insecticides for the application in pest biocontrol [10,11]. The virulence and pathological process of *M. robertsii* against pests have been paid extensive attention [12,13].

*Monochamus alternatus* Hope (Coleoptera: Cerambycidae) is mainly distributed in China, Korea, Japan, and several southeast Asian countries [14,15]. The *M. alternatus* is an important forest pest, and it causes serious harm to trees by feeding on woods directly or toting the phytopathogens indirectly. It even dominates significant forest collapses [16]. Among them, a particularly important point is that it can spread pine wilt disease [17]. Pine wilt disease, caused by the pine wood nematode, is a destructive forest disease, and it seriously endangers forestry safety all over the world [18]. For example, in 2017, the total damaged area of pine wilt disease reached 85,524 hm^2^ in China, and it caused economic losses of about USD 2.94 billion [19]. At present, the main methods of managing *M. alternatus* are chemical pesticides such as neonicotinoids, thiamethoxams, thiacloprids and fenitrothions [20,21]. However, these chemical pesticides are harmful to the environment. And chemical pesticides can lead to insect resistance. For instance, a worldwide insect pest, *Spodoptera frugiperda,* was found to have developed resistance to traditional pesticides like pyrethroids [22]. But in recent years, it was also found to be resistant to new pesticides like spinosad [23]. So natural microbial pesticides have been investigated as alternatives to chemical pesticides [24]. In fact, entomopathogenic fungi could pose a threat to non-target insect species, such as bees and soil microarthropods [25,26]. But the adverse influences were slight [27]. Some species of beneficial insects have even been used to spread entomopathogenic fungi in some studies [28,29].

Currently, there have been many studyies about using various *Metarhizium* strains to control *M. alternatus* such as *Metarhizium anisopliae* JEF-197 [30] and *Metarhizium anisopliae* F52 [31]. But the biological control strategies require continuous strains isolated from nature [32]; more entomopathogenic fungi need to be isolated and used in the pest management. In addition, entomopathogenic fungi intrude the host body mostly through the penetration of the cuticle, proliferation within the host body, and dissolution of host tissue; finally, the host dies [33,34]. However, the physiological changes of *M. alternatus* after infection, including symptom observation and histopathological observation, have received little attention. In this study, the *M. robertsii* strain GQH6 isolated from the Loess Plateau was identified morphologically and molecularly, and it was found high virulence against the third-instar larvae of *M. alternatus*. Then, the symptom observation and histopathological observation after infection were carried out to better understand the pathological process of the *M. robertsii* against the third-instar larvae of *M. alternatus*. In conclusion, this study aimed to provide a new strain and theoretical basis for the biological control of *M. alternatus*.

## 2. Materials and Methods

### 2.1. Isolation and Identification of the Metarhizium robertsii Strain GQH6

#### 2.1.1. Specimen Collection and Isolation

The *M. robertsii* strain GQH6 was isolated by the insect bait method [35] from soil samples collected in the Loess Plateau. The insect bait *Tenebrio molitor* (Coleoptera: Tenebrionidae), raised at LIRR (Lab of Insect Relative Resource, College of Plant Protection, Northwest A & F University, Shaanxi Province, Yangling, China), was used. And the soil samples were collected from the Geqiuhe Village, Yuyang District, Yulin City, Shaanxi Province in June 2021 (109°36.339′ E, 38°42.055′ N). The location belongs to the Mu Us desert in the Loess Plateau. After isolation, the *M. robertsii* strain GQH6 were stored at 4 °C in LIRR.

#### 2.1.2. Morphological Identification of the *Metarhizium robertsii* Strain GQH6

The colony morphology of the *M. robertsii* strain GQH6 were observed after being cultured on the SDAY medium (10 g/L peptone, 40 g/L D-glucose anhydrous, 10 g/L yeast extract, 20 g/L agar, 1 L steriled water) in an artificial incubator at 25 ± 2 °C, 60 ± 5% RH, 12:12 L:D photoperiod for 14 days. The obverse and reverse of the *M. robertsii* strain GQH6 on SDAY medium were observed. And the morphological characteristics, including the characteristics of mycelia and conidia of the *M. robertsii* strain GQH6, were also observed on a microscope (Ningbo Sunny Instruments Co., Ltd., Zhejiang, Ningbo, China).

#### 2.1.3. Molecular Identification of the *Metarhizium robertsii* Strain GQH6

The genomic DNA of the *M. robertsii* strain GQH6 was extracted using the method described by Aljanabi [36] with some modifications. The purified genomic DNA, as a template, was used to amplify the target genes (ITS and EF1α) by polymerase chain reaction (PCR). The internal transcribed spacer (ITS) regions of rDNA were amplified with the primer pair ITS1F (5′-TCCGTAGGTGAACCTGCGG-3′) and ITS4R (5′-TCCTCCGCTTATTGATATGC-3′) [37]. And the elongation factor 1-alpha (EF1α) regions were amplified with the primer pair EF1α-EF (5′-GCTCCYGGHCAYCGTGAYTTYAT-3′) and EF1α-ER (5′-ATGACACCRACRGCRACRGTYTG-3′) [38]. Then, the sequencing was carried out at Sangon Biotech (Shanghai, China). The sequences were deposited into the NCBI GenBank Database (NCBI, https://www.ncbi.nlm.nih.gov/ accessed on 17 January 2024). And all sequences were compared with already-published sequences using the BLAST tool from the NCBI GenBank Database [39]. After that, we used MEGA X to align each sequence. Then, a phylogenetic tree was formed based on the maximum likelihood (ML) method in PhyloSuite v1.2.1 [40,41,42].

### 2.2. Bioassay of the Metarhizium robertsii Strain GQH6

#### 2.2.1. Experimental Insects

The insect tested in this study was the third-instar larvae of *Monochamus alternatus* Hope (Coleoptera: Cerambycidae). The larvae were obtained commercially and reared on sawdust in 24-well plastic boxes ventilated with several holes at 25 ± 2 °C, 60 ± 5% RH with completely darkness. Third-instar larvae with good condition of *M. alternatus* were selected for this bioassay.

#### 2.2.2. Preparation of Fungal Suspension

Before the bioassay, the *M. robertsii* strain GQH6 was cultured by the one-quarter-strength SDAY medium (2.5 g/L peptone, 10 g/L D-glucose anhydrous, 2.5 g/L yeast extract, 20 g/L agar, 1 L sterile water) in an artificial incubator at 25 ± 2 °C, 60 ± 5% RH, and a 12:12 L:D photoperiod for 14 days to produce enough conidia. The conidia were scraped by the sterilized wood chips, and then they were transferred into 20 mL of sterile water containing 1% glycerin and 0.05% Tween-80. The suspension was adequately vortexed by a vortex oscillator. The conidia concentration of suspension were determined using a hemocytometer under 10× or 40× magnification on the microscope (ECLIPSE TE2000-S, Nikon, Tokyo, Japan). Then, the suspension was diluted into the concentrations of 10^9^, 10^8^, 10^7^, 10^6^ and 10^5^ conidia/mL, and 20–40 mL for each concentration. The sterilized water containing 1% glycerin and 0.05% Tween-80 was for the control group.

#### 2.2.3. Bioassay

The 15 healthy larvae of third instar were selected for one biological replicate of each concentration. There were three biological replicates for each concentration. In total, in three biological replicates of each concentration, the 45 third-instar larvae of *M. alternatus* were treated. The impregnation was the method used for the bioassay in this study. For one larva, it was placed into the suspension for 3–5 s, so that it was fully covered with the suspension, and then the excess water in its body was removed by sterilized filter paper. After infection, they were transferred to the 24-well plastic boxes immediately. Then, all the 24-well plastic boxes were stored at an artificial incubator at 25 ± 2 °C, 60 ± 5% RH in completely darkness. The mortality of larvae was observed and recorded for 21 consecutive days.

### 2.3. Symptom Observations and Histopathological Observations

After being infected by the *M. robertsii* strain GQH6, the symptoms of infected larvae were observed every day. In addition, on 0, 2, 4, 6, 8 days after infection by the suspension of 10^9^ conidia/mL, three infected larvae with typical symptoms were picked, and they were individually put into a 5 mL tube filled with 4% paraformaldehyde solution. Then, they were sent to Y&KBio (Xi’an, China) for making tissue sections after paraffin embedding, paraffin sectioning and HE staining. After that, the tissue sections were observed under a microscope (Ningbo Sunny Instruments Co., Ltd., Zhejiang, Ningbo, China) to accomplish histopathological observation.

### 2.4. Statistical Analysis

For all experiments, we performed three biological replicates. The calculation of corrected mortality (%) was as follows: corrected mortality (%) = (mortality of fungal infected group − mortality of control group)/(1 − mortality of control group) × 100%. The lethal concentrations (LC50 and LC90), were calculated using GraphPad Prism 8.0.2. And the median lethal time (LT50 and LT90) was calculated with probit analysis (IBM SPSS Statistics 27). The *p* < 0.05 was considered statistically significant. All figures were produced by GraphPad Prism 8.0.2.

## 3. Results

### 3.1. Identification of the Metarhizium robertsii Strain GQH6

The *M. robertsii* strain GQH6 was molecularly identified by a phylogenetic tree using the ribosomal internal transcribed spacer region (ITS) and the elongation factor 1-alpha (EF1α) regions (Figure 1). The strain GQH6 and *Metarhizium robertsii ARSEF 727* were clustered in the same branch. And the accession numbers of the NCBI Database were submitted (ITS: PP140918; EF1α: PP125304). After being cultured for 14 days on the SDAY medium, the colony diameter of strain GQH6 was 4.7 cm (Figure 2a). On the obverse of the colony, the center was reseda green and the edge was white (Figure 2a). On the reverse of the colony, the center was pale yellow and the edge was white (Figure 2b). The hyphae were achromic and linear (Figure 2d), and the conidia were achromic and elliptic (Figure 2c). Using molecular identification combined with morphological identification, we discovered the strain GQH6 was *Metarhizium robertsii*.

### 3.2. Virulence of the Metarhizium robertsii Strain GQH6 against Monochamus alternatus Larvae

The corrected mortality (%), number of dead larvae, LC50/LC90 and LT50/LT90, were, respectively, recorded and calculated for virulence of the *M. robertsii* strain GQH6. With the increase in concentration, the corrected mortality of larvae rose gradually. The corrected mortality caused by the *M. robertsii* strain GQH6 could reach 100% at high concentration (10^9^ conidia/mL and 10^8^ conidia/mL), while the corrected mortality was 88.89 ± 5.88% at 10^7^ conidia/mL, 44.44 ± 5.88% at 10^6^ conidia/mL, and 17.78 ± 2.22% at 10^5^ conidia/mL (Figure 3a). At 10^9^ conidia/mL, the larvae began to die after two days of infection, and all the larvae had died by the seventh day. And at 10^8^ conidia/mL, the larvae also began to die after two days, and all the larvae had died by the ninth day. At lower concentrations (10^7^ conidia/mL, 10^6^ conidia/mL and 10^5^ conidia/mL), the larvae began to die later and the mortality was less than 100% (Figure 3b). The *M. robertsii* strain GQH6 had high virulence against the third-instar larvae of *M. alternatus*, with the LC50 being 1.93 × 10^6^ conidia/mL and the LC90 being 1.35 × 10^7^ conidia/mL (Table 1). Similarly, with the increase in concentration, the LT50 and LT90 of larvae reduced. At the highest concentration (10^9^ conidia/mL), the LT50 was 3.96 days, and the LT90 was 5.45 days (Table 2), and showed high virulence. And at 10^8^ conidia/mL, the LT50 was 4.99 days, and the LT90 was 7.78 days. It was also at a good level. In conclusion, the results indicated that the strain GQH6 of *M. robertsii* showed high virulence against the third-instar larvae of *M. alternatus*.

### 3.3. Symptom Observation of Infected Larvae

The symptoms of third-instar larvae of *M. alternatus* infected by the *M. robertsii* strain GQH6 were obvious (Figure 4). The symptoms of larvae treated with high concentration of conidia suspension appeared earlier than those treated with low concentration. Here, a batch of larvae, which died earliest, were taken as samples and analyzed. At 10^9^ conidia/mL, after being injected by strain GQH6 for one day, the motion and epidermis of larvae were not different with non-infected larvae (Figure 4a). Then, the dark spots appeared and increased in number (Figure 4b–d), and the motion of the larvae became phlegmatic (Figure 4c,d). The larvae began to die after two days of inoculation. At the fourth day, the white mycelia appeared on the surface of the dead larvae (Figure 4e); the larva bodies became stiff. Similarly, at the fourth day, the living larvae were basically inactive and covered with massive dark spots (Figure 4d). And then, the surface of the larvae was covered with white mycelia. By the sixth day, green conidia appeared on the surface of the larvae (Figure 4f) and gradually increased in number. Finally, the larva bodies became green (surrounded by conidia at the eighth day) (Figure 4g). After the appearance of the white mycelia, the larva bodies were always stiff (Figure 4e–g).

### 3.4. Histopathological Observation of Infected Larvae

The histopathological observation of the third-instar larvae of *M. alternatus* infected by the *M. robertsii* strain GQH6 was carried out through the observation of tissue slices. The muscle tissue (MT), adipose tissue (AT), and midgut (Mg) were the focus of observation (Figure 5, Figure 6, Figure 7 and Figure 8).

At 0 day after infection, the muscle tissue was compact, and there were no hyphae and conidia (Figure 5a). At the second day, the hyphae (Hy) and conidia (Co) appeared around the muscle tissue (Figure 5b). At the fourth day, the hyphae and conidia became more, and the congregation of hematocytes (H) appeared (Figure 5c). At the sixth day, there were large numbers of hyphae attached to the muscle tissue, and the muscle tissue became loose (Figure 5d). At the eighth day, the muscle tissue was almost broken down, and the haemocoel was filled with hyphae and conidia (Figure 5e).

Regarding the adipose tissue, there were no obvious changes in the first 2 days (Figure 6a,b); only a little hyphae had appeared (Figure 6b). Then, the hyphae increased, and the adipose tissue was destroyed (Figure 6c). At the sixth day, the adipose tissue was consumed by hyphae, and the adipose tissue became blurred (Figure 6d). Finally, at the eighth day, the adipose tissue was completely blurred, and the adipose tissue was filled with hyphae and conidia (Figure 6e).

Inside the midgut, there were no obvious changes in the first 4 days (Figure 7a–c), except hyphae appearing around the midgut (Figure 7c). The hyphae and conidia passed through the body wall, muscle tissue and adipose tissue of larvae, and then the hyphae and conidia continued to proliferate and reach the midgut. The symptoms of the midgut appeared later than those of muscle tissue and adipose tissue. At the sixth day, the haemocoel around the midgut was filled with hyphae and conidia, and the midgut wall was destroyed; a large number of hyphae had appeared inside the midgut (Figure 7d). Finally, at eight days, the midgut structure became completely blurred (Figure 7e).

Overall, after being infected by the *M. robertsii* strain GQH6, there were enormous changes in the body of the *M. alternatus* larvae (Figure 8). The hyphae and conidia continued to proliferate from the outside to the inside. At the fourth day, the experimented larvae died because of the hyphae growth and tissue destruction (Figure 8c). After that, the larvae body became stiff. As showed in Figure 8d, the larvae body filled with hyphae. It induced stiffness of the larvae. And the extrusion of hyphae continued to destroy the tissue (Figure 8c,d), then the body of larvae completely filled with hyphae and conidia (Figure 8d,e).

## 4. Discussion

In this study, the entomopathogenic fungi strain was identified as *Metarhizium robertsii* by molecular identification and morphological identification. The internal transcribed spacer (ITS) regions of rDNA and the elongation factor 1-alpha (EF1α) regions were frequently used for the molecular identification of fungi [37,38]. Identification of multiple genes (ITS and EF1α) improved the accuracy of molecular identification in this study. Using a phylogenetic tree, the strain GQH6 was identified as *Metarhizium robertsii*. The morphological characteristics of the fungi, such as its macro-morphology and micro-morphology, were also used to help to identify the fungal species [43]. The colony color on the medium of *Metarhizium* strains was reseda green [44]. Through microscopic observation, the conidia of *Metarhizium* were found to be oval [45]. This is consistent with the morphological characteristics of *Metarhizium*.

The species of entomopathogenic fungal, which were found to be able to infect the *M. alternatus*, were mainly *Beauveria* and *Metarhizium* [17,46]. Kim et al. [47] found that the virulence of *Metarhizium anisopliae* JEF-279 against *M. alternatus* was high. And the *Metarhizium* was also found to be highly virulent against *M. alternatus* under field conditions [48]. Gebremariam et al. [49] defined that the LT50 value of entomopathogenic fungal strains <5 days was high virulence, between 5–6 days was moderate virulence and >6 days was low virulence. The LT50 of the *M. robertsii* strain GQH6 was 3.96 days at 10^9^ conidia/mL, and 4.99 days at 10^8^ conidia/mL. It showed high virulence against the third-instar larvae of *M. alternatus*. There are many factors determining the virulence of fungal strains. For example, fungal conidia can secrete a variety of cuticle-degrading enzymes to help to penetrate an insect epidermis, and the activity and quantity of cuticle-degrading enzymes also determine the virulence of the fungi [50]. Some entomopathogenic fungi can secrete secondary metabolites, which can help the fungi to overcome the immune system of insects [51]. The factors affecting the high virulence of the *M. robertsii* strain GQH6 against *M. alternatus* were still unclear and need further research to confirm these hypotheses. In addition, the corrected mortality of the *M. robertsii* strain GQH6 against the third-instar larvae of *M. alternatus* was 100% at 10^9^ and 10^8^ conidia/mL, 88.89 ± 5.88% at 10^7^ conidia/mL. And the LC50 of the strain GQH6 was 1.93 × 10^6^; the LC90 was 1.35 × 10^7^. These virulence indices showed high virulence against the *M. alternatus* larvae.

The symptom observation and histopathological observation were also necessary parts for verifying the virulence of the entomopathogenic fungal strain [47,52]. After infected by the strain GQH6, dark spots appeared on the epidermis of larvae, and increased. It was reported that melanization (dark spots on the epidermis of larvae after infection) is a reaction of the infected larvae to, for example, enhance cuticle pigmentation, heal its wound, and activate its innate immune system [53]. After the larvae were infected, there were many potent elicitors produced by fungi such as β-1,3-glucans [54]. This could cause a strong immune response of the larvae and it might be the reason of the first appearance of the dark spots. The motion of the larvae were found to be slow in this study. Kim et al. [47] proved that some proteins produced by entomopathogenic fungi could cause flaccid paralysis and tissue damage. It is speculated to be a major element of the insecticidal mechanism. With the fungus gradually proliferated in the larvae body, it eventually destroyed the tissue through toxins [12,13] and mycelium extrusion (Figure 5, Figure 6, Figure 7 and Figure 8) [47], and then it killed the larvae. The dead larvae were also found to be stiff in this study. Through histopathological observation, we conjectured that the reason was that the mycelia grew continuously, and finally the mycelia filled the larvae body. Entomopathogenic fungi can infect the insects by penetration of the cuticle, and ingestion is not necessary [55]. This helps to improve the virulence. In this study, the method of infection was immersion, not through the digestive system of the *M. alternatus* larvae (Figure 7a–c). After infection by the *M. robertsii* strain GQH6, the histopathological changes to the larvae were obvious. The main symptoms were muscle tissue rupture, adipose tissue fracture and midgut disintegration (Figure 5, Figure 6, Figure 7 and Figure 8). Through observation of tissue sections, mycelia growth and tissue damage were found (Figure 5, Figure 6, Figure 7 and Figure 8). As for which genes or substances played a decisive role in destroying tissues, this needs to be further researched [52]. It was roughly same as the histopathological changes of *Beauveria bassiana* against Colorado potato beetle larvae [56]. These results indicated that the *M. robertsii* strain GQH6 isolated from the Loess Plateau was highly virulent against the third-instar larvae of *M. alternatus*, not only by the virulence indices, but also by symptom observation and histopathological observation. And this study also provided detailed descriptions for the pathological process of the *M. robertsii* against the *M.alternatue* larvae. In addition, “native” entomopathogenic fungi are more adaptable to environmental conditions [57]. So the native entomopathogenic fungal strains can increase the effectiveness of pest control [58], while non-native entomopathogenic fungal strains decrease the virulence and even bring some ecological risks [59]. Therefore, entomopathogenic fungi isolated from their natural environment are of great significance to biological control. The *M. robertsii* strain GQH6, isolated from the Loess Plateau, was highly virulent against the third-instar larvae of *M. alternatus*. In the future, if an outbreak of *M. alternatus* occurs in the Loess Plateau, the *M. robertsii* strain GQH6 will be a good choice for biological control. Even though our study was carried out in laboratory conditions, our previous study developed a new method to transmit entomopathogenic fungi by a vector mite into the frass holes of long-horn beetles [60]. Therefore, it is possible to transmit the spores of the *M. robertsii* strain GQH6 to *M. alternatus* larvae by the new method described above. On the other hand, some entomopathogenic fungi strains were found to be highly virulent against *Buraphelenchus xylophilus* [61]. In the future, the potential of the *M. robertsii* strain GQH6 to control *Buraphelenchus xylophilus* needs to be further studied.

## Figures and Tables

**Figure 1 microorganisms-12-00514-f001:**
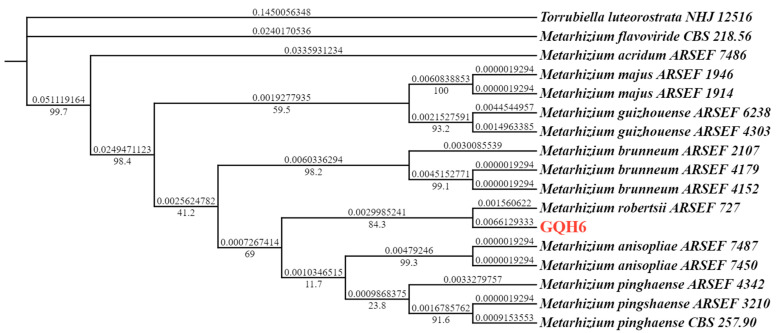
Phylogenetic tree based on ITS and EF1α regions by PhyloSuite v1.2.1. The phylogenetic tree was formed based on the maximum likelihood (ML) method in PhyloSuite v1.2.1 and rooted by *Torrubiella luteorostrata NHJ12516* as outgroup.

**Figure 2 microorganisms-12-00514-f002:**
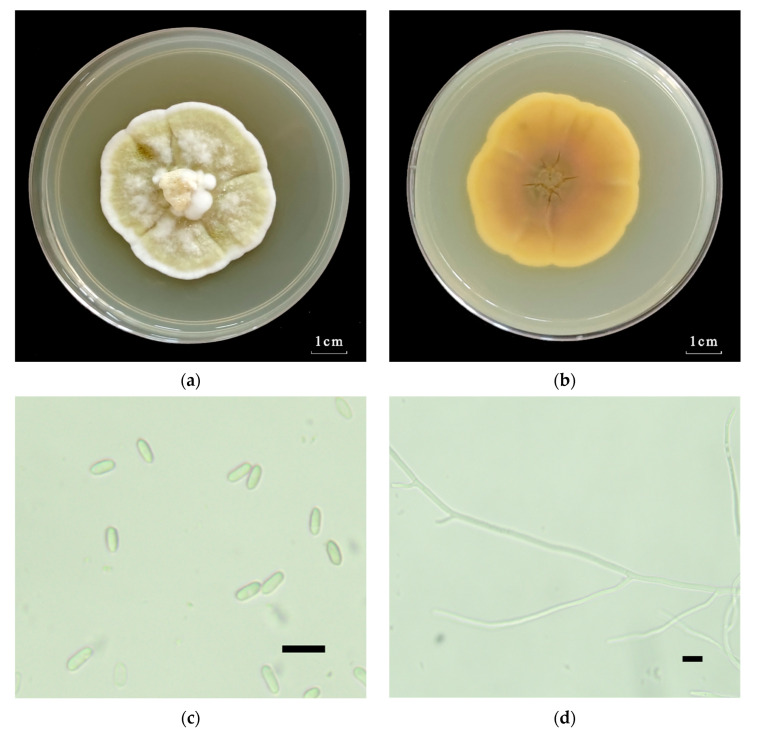
The morphological feature of the *M. robertsii* strain GQH6. (**a**) The obverse of the *M. robertsii* strain GQH6 on SDAY medium; (**b**) the reverse of the *M. robertsii* strain GQH6 on SDAY medium; (**c**) the feature of conidia (Bars = 10 µm); (**d**) the feature of hyphae (Bars = 10 µm).

**Figure 3 microorganisms-12-00514-f003:**
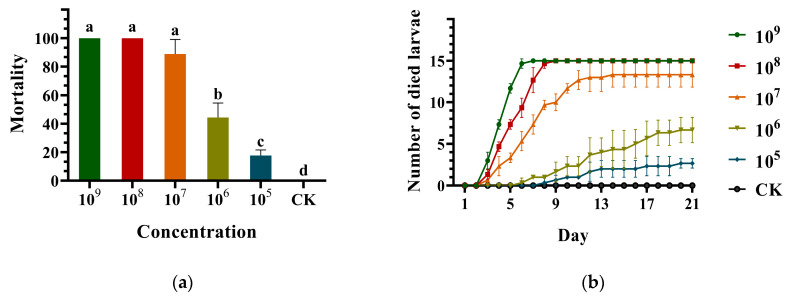
The histogram of corrected mortality (%) at different concentrations and the line chart of the number of dead larvae everyday at different concentrations. (**a**) The histogram of corrected mortality (%) at different concentrations; (**b**) the line chart of the number of dead larvae everyday at different concentrations. The third-instar larvae of *M. alternatus* were treated by the conidia suspension of the *M. robertsii* strain GQH6 with different concentrations (10^9^, 10^8^, 10^7^, 10^6^ and 10^5^). After that, the number of dead larvae was recorded everyday until 21 days later. The bioassay was conducted with three replicates (15 larvae/replicate). Different letters (a, b, c, d) indicated statistical differences between different concentrations (*p* < 0.05).

**Figure 4 microorganisms-12-00514-f004:**
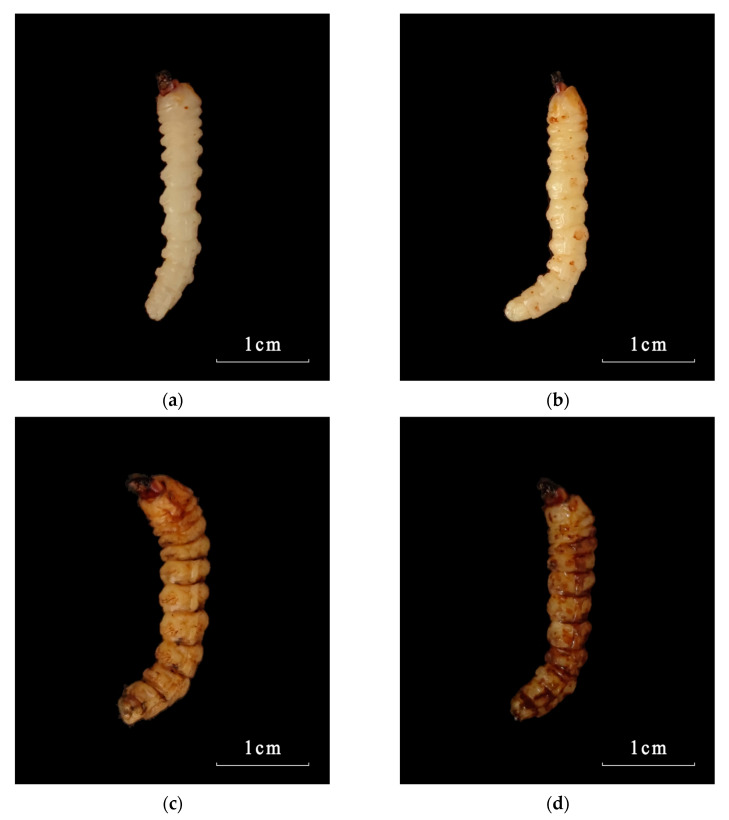
The symptoms of *M. alternaue* larvae after infection by the *M. robertsii* strain GQH6. (**a**–**d**) The symptoms of infected larvae in the first three days; (**e**) the symptoms of infected larvae at the fourth day; (**f**) the symptoms of infected larvae at the sixth day; (**g**) the symptoms of infected larvae at the eighth day.

**Figure 5 microorganisms-12-00514-f005:**
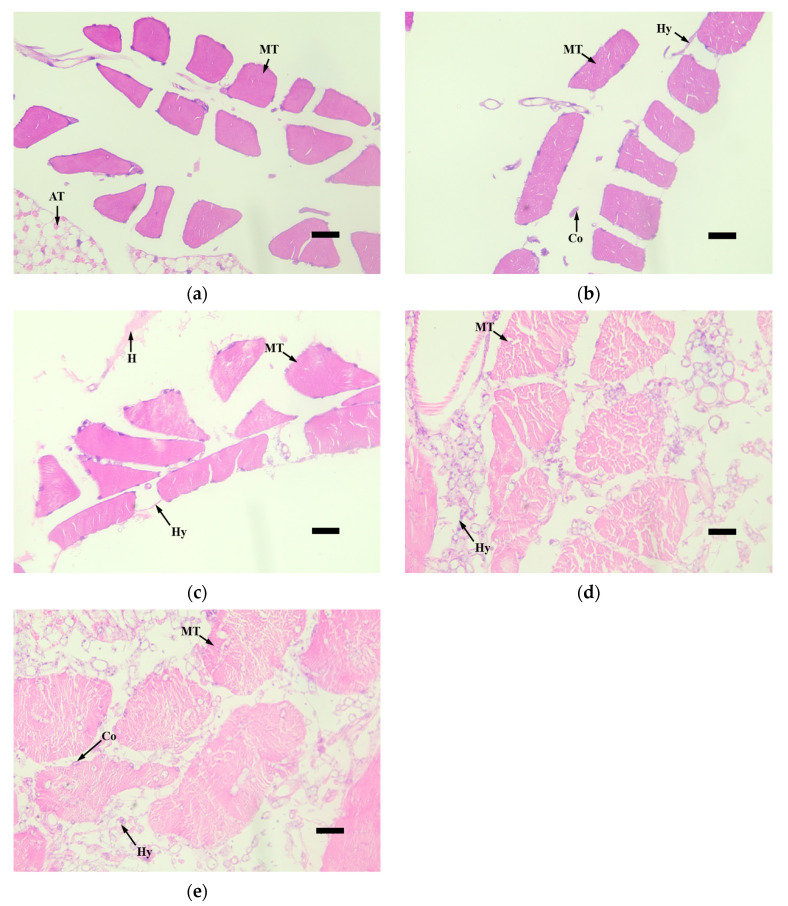
The tissue slices of muscle tissue (MT) of *M.alternaue* larvae after infection by the *M. robertsii* strain GQH6. (**a**) The histopathological observation of muscle tissue at 0 days; (**b**) the histopathological observation of muscle tissue at second days; (**c**) the histopathological observation of muscle tissue at fourth days; (**d**) the histopathological observation of muscle tissue at sixth days; and (**e**) the histopathological observation of muscle tissue at eighth days. MT: muscle tissue; AT: adipose tissue; Hy: hyphae; Co: conidia; H: congregation of hematocytes. Bars = 10 µm.

**Figure 6 microorganisms-12-00514-f006:**
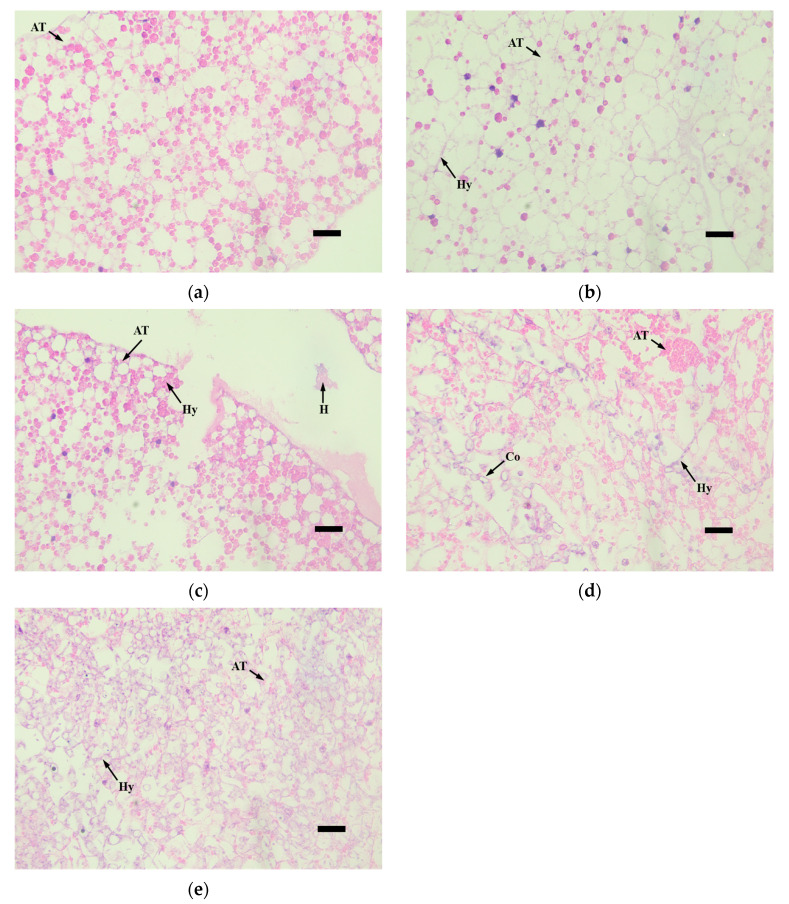
The tissue slices of adipose tissue (AT) of *M.alternaue* larvae after infection by the *M. robertsii* strain GQH6. (**a**) The histopathological observation of adipose tissue at 0 days; (**b**) the histopathological observation of adipose tissue at second days; (**c**) the histopathological observation of adipose tissue at fourth days; (**d**) the histopathological observation of adipose tissue at sixth days; (**e**) the histopathological observation of adipose tissue at eighth days. AT: adipose tissue; Hy: hyphae; Co: conidia; H: congregation of hematocytes. Bars = 10 µm.

**Figure 7 microorganisms-12-00514-f007:**
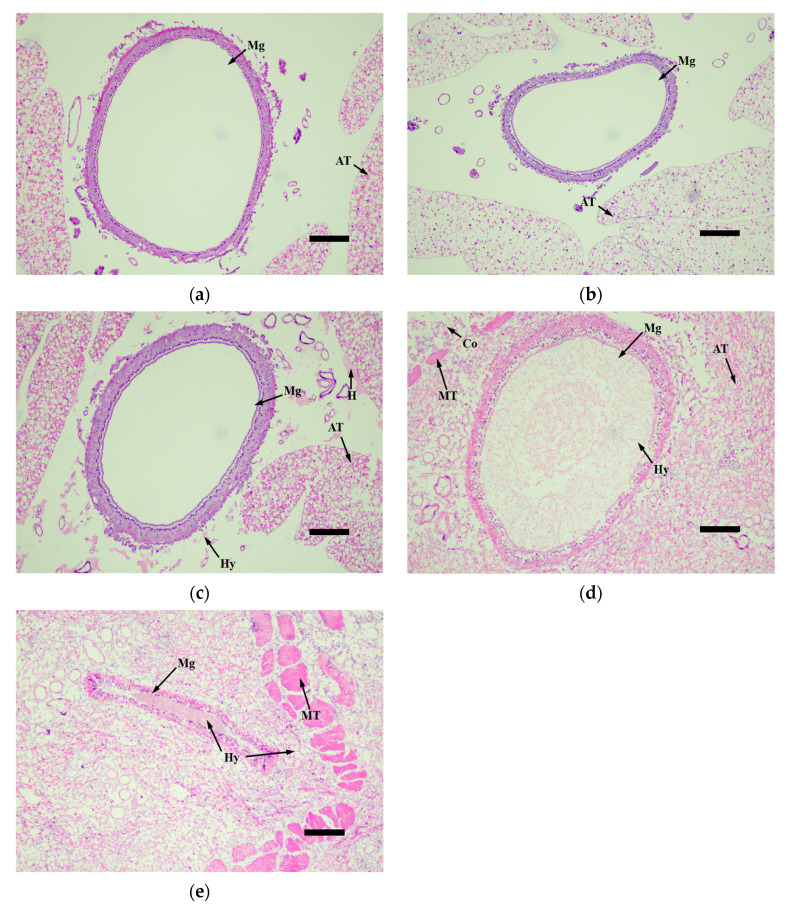
The tissue slices of the midgut (Mg) of *M.alternaue* larvae after infection by the *M. robertsii* strain GQH6. (**a**) The histopathological observation of the midgut at 0 days; (**b**) the histopathological observation of the midgut at second days; (**c**) the histopathological observation of the midgut at fourth days; (**d**) the histopathological observation of midgut at sixth days; (**e**) the histopathological observation of the midgut at eighth days. Mg: midgut; MT: muscle tissue; AT: adipose tissue; Hy: hyphae; Co: conidia; H: congregation of hematocytes. Bars = 100 µm.

**Figure 8 microorganisms-12-00514-f008:**
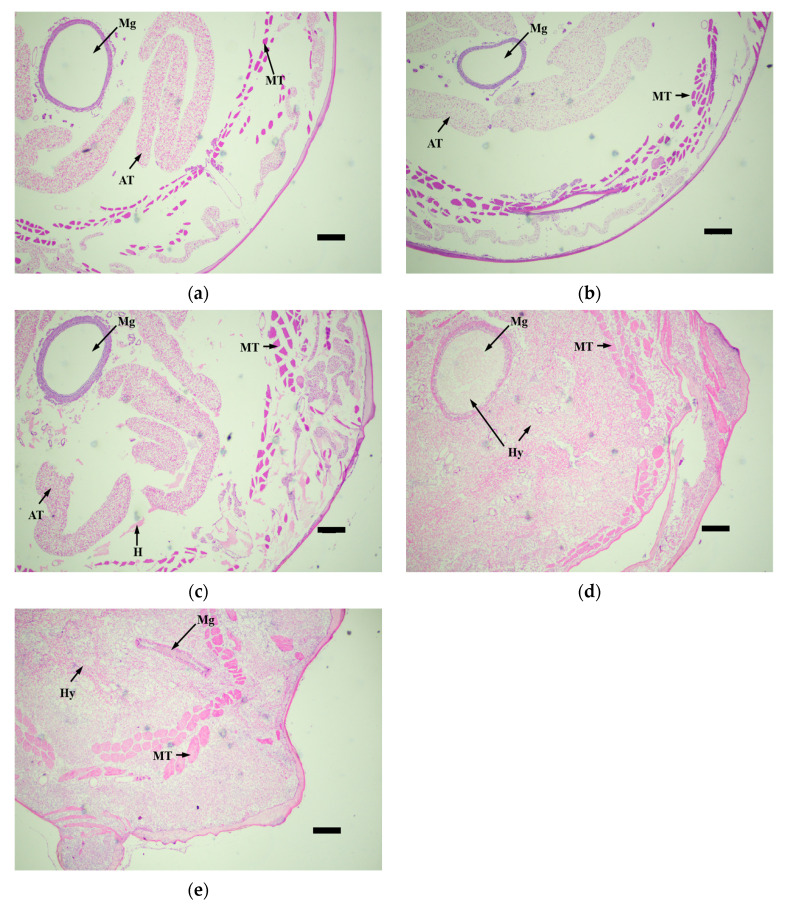
The tissue slices of muscle tissue (MT), adipose tissue (AT) and the midgut (Mg) of *M.alternaue* larvae after infection by the *M. robertsii* strain GQH6. (**a**) The histopathological observation of muscle tissue, adipose tissue and the midgut at 0 days; (**b**) the histopathological observation of muscle tissue, adipose tissue and the midgut at second days; (**c**) the histopathological observation of muscle tissue, adipose tissue and the midgut at fourth days; (**d**) the histopathological observation of muscle tissue, adipose tissue and the midgut at sixth days; and (**e**) the histopathological observation of muscle tissue, adipose tissue and the midgut at eighth days. Mg: midgut; MT: muscle tissue; AT: adipose tissue; Hy: hyphae; Co: conidia; H: congregation of hematocytes. Bars = 100 µm.

**Table 1 microorganisms-12-00514-t001:** The LC50 and LC90 of the *M. robertsii* strain GQH6 against the third-instar larvae of *M. alternatus*.

Fungal Number	LC50 (Spore/mL)	95% Confidence Interval	LC90 (Spore/mL)	95% Confidence Interval
GQH6	1.93 × 10^6^	1.40 × 10^6^–2.69 × 10^6^	1.35 × 10^7^	3.85 × 10^6^–4.41 × 10^7^

**Table 2 microorganisms-12-00514-t002:** The LT50 and LT90 of the *M. robertsii* strain GQH6 against the third-instar larvae of *M. alternatus*.

Fungal Number	Conidial Concentration	LT50 (Days)	95% Confidence Interval	LT90 (Days)	95% Confidence Interval
GQH6	10^9^	3.96	3.85–4.08	5.45	4.83–6.43
	10^8^	4.99	4.79–5.18	7.78	5.66–12.99
	10^7^	6.49	6.19–6.78	11.37	10.12–13.18
	10^6^	11.83	10.99–12.66	-	-

## Data Availability

The sequences of the *Metarhizium robertsii* strain GQH6 are available in the Genback datable.

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
