# Peer review of "A New Strain of Metarhizium robertsii Isolated from Loess Plateau and Its Virulence and Pathological Characteristics against Monochamus alternatus"

_microorganisms, 2024, doi:10.3390/microorganisms12030514_

Round 1
Reviewer 1 Report
Comments and Suggestions for Authors
Dear authors,
Will the introduction of larger quantities of the product into the environment pose a threat to other insect species? Could beneficial ants, bees or other pollinators be jeopardised? How will the long-term use of the biological product affect the biodiversity of the ecosystem? Can the isolated strain from the Loess Plateau be used only there or can it be safely transferred to other areas in China or abroad?
What does the effectiveness of the product depend on? On the temperature? On the humidity? Add how important Monochamus alternaris is economically in China to justify the need for eradication. Is it involved in the transmission of the harmful nematode Buraphelenchus xylophilus, which kills conifers? Is it mainly used to protect pine stands? How are the preparations applied? Spraying from the ground or from the air?
The authors use good English and the article reads well.
Reviewer 2 Report
Comments and Suggestions for Authors
Dear authors,
Your manuscript is quite good in terms of subject matter, which is obviously of interest and future.
However, some improvement suggestions are needed to increase the value but also to strengthen the wide-scale applicability:
Line 55-54: Related to: <However, these chemical pesticides are harmful to the environment, otherwise, chemical pesticides can lead insect resistance.>
The phrase is too general, not providing a clear justification. These pesticides are being phased out in some parts of the world, so you should provide more details and give weight to the statement. And you should attribute 1-2 sources that say this.
​ Line 59-60: < Currently, there were many researches about using various Metarhizium strains to
control M.alternatus >
Please exemplify or clarify which strains were tested to separate things and see what you bring new to what is already published.
Line 93-94: Figure 1 is not necessary, in my opinion it can be removed, the 2 maps are not relevant as long as the external climatic factors have not been considered and the research locations are not compared. The experiment was mostly carried out in the laboratory, so far the identification coordinates are already described.
Line 384: Provide a concluding sentence that provides guarantees on the broad applicability of the results, specifically why they are and what they are applicable to.
Kind regards,
R
